# PDX-1: A Promising Therapeutic Target to Reverse Diabetes

**Yanjiao Zhang** [1], **Xinyi Fang** [1,2], **Jiahua Wei** [3], **Runyu Miao** [1,2], **Haoran Wu** [2], **Kaile Ma** [1] and **Jiaxing Tian** [1,*]

1   Institute of Metabolic Diseases, Guang'anmen Hospital, China Academy of Chinese Medical Sciences, Beijing 100053, China
2   Graduate College, Beijing University of Chinese Medicine, Beijing 100029, China
3   Graduate College, Changchun University of Chinese Medicine, Changchun 130117, China
*   Correspondence: tina_yai@126.com

**Abstract:** The pancreatic duodenum homeobox-1 (*PDX-1*) is a transcription factor encoded by a Hox-like homeodomain gene that plays a crucial role in pancreatic development, β-cell differentiation, and the maintenance of mature β-cell functions. Research on the relationship between *PDX-1* and diabetes has gained much attention because of the increasing prevalence of diabetes melitus (DM). Recent studies have shown that the overexpression of *PDX-1* regulates pancreatic development and promotes β-cell differentiation and insulin secretion. It also plays a vital role in cell remodeling, gene editing, and drug development. Conversely, the absence of *PDX-1* increases susceptibility to DM. Therefore, in this review, we summarized the role of *PDX-1* in pancreatic development and the pathogenesis of DM. A better understanding of *PDX-1* will deepen our knowledge of the pathophysiology of DM and provide a scientific basis for exploring *PDX-1* as a potential target for treating diabetes.

**Keywords:** *PDX-1*; diabetes mellitus; reversal; transcription factor; β-cell

## 1. Introduction

Diabetes mellitus (DM) is a metabolic disorder with an increasing prevalence worldwide. It is characterized by chronic hyperglycemia and disturbances in protein, carbohydrate, and fat metabolism due to insulin resistance (IR) and/or insulin secretion deficiency [1]. Approximately one billion people suffer from chronic hyperglycemia globally, a major public health problem. The International Diabetes Federation estimates that 10.5% (537 million) of adults aged 20–79 years are currently living with DM, and this prevalence is expected to increase to 11.3% (643 million) by 2030 and 12.2% (783 million) by 2045. With 1.541 million adults having impaired glucose tolerance (IGT), their risk for type 2 diabetes is increased [2]. Diabetes causes a series of complications, including blindness, renal failure, stroke, and coronary artery disease, resulting in a huge medical burden on society [3]. Furthermore, diabetes costs at least 966 billion dollars in health expenditure, a 316% increase over the last 15 years [2].

DM is divided into type 1 diabetes (insulin-dependent diabetes mellitus (T1DM or IDDM)), type 2 diabetes (non-insulin-dependent diabetes mellitus (T2DM or NIDDM)), specific types of diabetes due to other causes, and gestational diabetes mellitus [4,5]. T2DM is the most common presentation of DM, accounting for approximately 90% of DM cases, whereas T1DM constitutes more than 5% [6,7]. T2DM is a chronic multisystem disease characterized by insulin resistance and elevated blood glucose levels [8]. It is the result of a complex interplay between genetic, epigenetic, and environmental factors [9,10]. However, its etiology and pathogenesis have not yet been fully elucidated. The traditional view is that diabetes can only be controlled and not cured. Authoritative guidelines and clinical diabetes research mostly focus on controlling blood glucose levels and improving complications. Blood glucose levels are controlled by promoting insulin secretion, enhancing insulin sensitivity, and promoting glucose absorption by other tissues outside the islets

of Langerhans [11,12]. With recent advances in diabetes research and medical technology progress, researchers have discovered new methods for preventing and treating diabetes with promising results. The terms "Diabetes reversal" and "Diabetes remission" have been used in scientific articles. They connote a glycosylated hemoglobin A1c (HbA1c) level of <6.5% (<48 mmol/mol) for at least 3 months without the usual glucose-lowering pharmacotherapy [13]. The reversal strategy, mechanism, and predictors are becoming increasingly clear with further research [14–17]. One of the possible mechanisms of diabetes reversal is that the removal of excess fat in the liver and pancreas can normalize hepatic glucose production and β-cell redifferentiation [14]. However, more studies are needed on the mechanism of diabetes reversal.

Studies on the role of *PDX-1* in reversing diabetes are on the increase. *PDX-1* regulates pancreatic development, β-cell differentiation, and the maintenance of mature β-cell function [18–24]. Non-β-cell can be reprogrammed by the transcription factors *PDX-1* and musculoaponeurotic fibrosarcoma oncogene family A (*MafA*) into functional β-cell, which secretes insulin to restore blood glucose levels [25,26]. In contrast, β-cell-specific removal of *PDX-1* resulted in pancreatic agenesis and severe hyperglycemia [27,28]. Thus, promoting *PDX-1* expression can be an effective strategy to ameliorate β-cell dysfunction and diabetes progression, making *PDX-1* a new target for developing anti-diabetic drugs. Therefore, in this review, we systematically summarized the roles of *PDX-1* in pancreatic development and function maintenance. Furthermore, we explored the application and prospects of *PDX-1* in diabetes research and treatment.

## 2. Gene Structure and Location, Protein Molecular Structure, Distribution, and Expression of *PDX-1*

The *PDX-1*, also known as *IUF-1* (insulin upstream factor 1), *IPF-1* (insulin promoter factor 1), *STF-1* (somatostatin transcription factor 1), and *IDX-1* (islet/duodenum homeobox-1), is a member of the homeodomain (HD)-containing transcription factor family and was first found in Xenopus laevis [18,29–32]. It plays a key role in the genesis, development, and maturation of the pancreas and is also one of the factors necessary for maintaining normal pancreatic islet function.

### 2.1. Localization and Molecular Structure of PDX-1

Rat and mouse *PDX-1* genes are localized on chromosomes 12 and 5, respectively, whereas the human *PDX-1* gene is located on chromosome 13q12 (12.1) [33–36]. The human *PDX-1* gene is approximately 6 Kb long with two exons. The first exon encodes the NH2-terminal region and some HD, while the second encodes the remaining HD and the COOH-terminal domain [37]. (Figure 1) Three nuclease-hypersensitive sites were identified within the 5′-flanking region of the endogenous *PDX-1* gene: HSS1($-2560 \sim -1880$ bp), HSS2($-1330 \sim -800$ bp), and HSS3($-260 \sim +180$ bp) [38]. Among them, HSS1 is an important functional region of *PDX-1* gene transcription activation and includes four sub-regions, region I ($-2694 \sim -2561$ bp), region II ($-2139 \sim -1958$ bp), and region III ($-1879 \sim -1799$ bp). The fourth distal enhancer element is region IV, located between $-6200$ and $-5670$ bp [39–41]. Regions I and II endow endocrine cell expression, region III mediates embryonic pancreas-wide expression, and region IV endows pancreas β-cell-specific gene expression and enhancement of proximal enhancer activity [42]. In addition, *PDX-1* transcription is also regulated by factors acting upon conserved Area I and IV sequences [38]. The proximal and distal promoters contain four conserved regions: I, II, III, and IV, which bind to various transcription factors to regulate cell differentiation [40]. In a study, a child's pancreas did not develop (pancreatic agenesis) because the child was homozygous for an inactivating cytosine deletion in the protein-coding sequence of *PDX-1* (pro63fsdelc), which indicated that *PDX-1* might be associated with Type 2 diabetes [43]. In addition, six novel *PDX-1* missense mutations (C18R, D76N, R197H, Q59L, G212R and P2390) were identified in patients with type 2 diabetes [44,45].

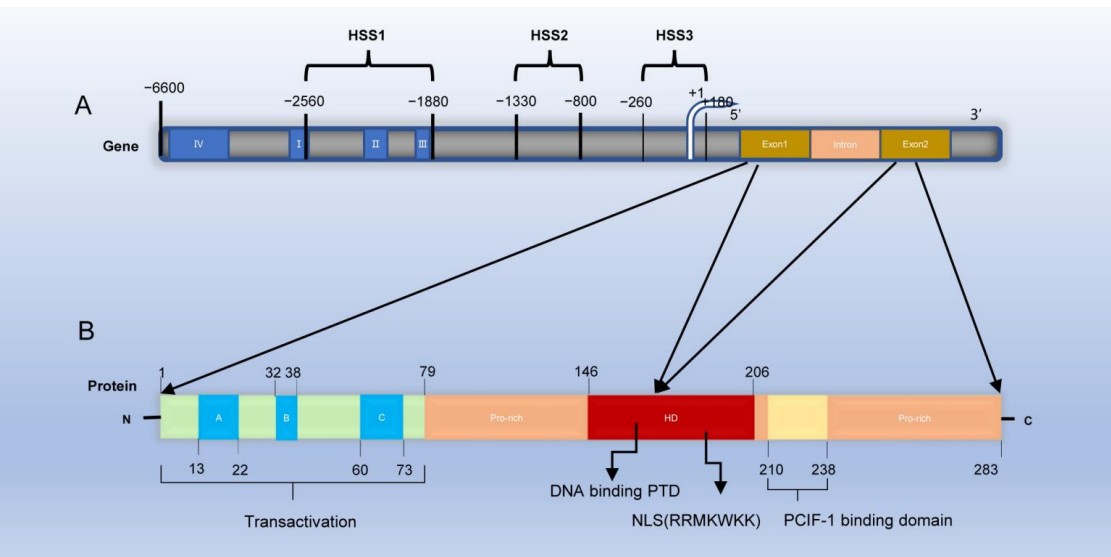

**Figure 1. Structure of *PDX-1* gene and protein.** (**A**) *PDX-1* gene contains two exons: exon 1 encodes the NH2-terminal domain and some homeodomain of PDX-1 protein, and exon 2 encodes the remaining homeodomain and COOH-terminal domain. Three nuclease-hypersensitive sites were identified within the 5′-flanking region of the endogenous *PDX-1* gene: HSS1 (−2560~−1880 bp), HSS2 (−1330~−800 bp) and HSS3 (−260~+180 bp). Among them, HSS1 is an important functional region of *PDX-1* gene transcription activation and includes four sub-regions, region I (−2694~−2561 bp), region II (−2139~−1958 bp), region III (−1879~−1799 bp), and region IV (−6200 and −5670 bp). (**B**) PDX-1 protein comprises 283 amino acids. The NH2-terminal is a proline-rich transcriptional activation domain, including 1~77 amino acids (amino acid AA or aa), which is composed of three highly conserved subdomains A, B, and C (A:13−22aa B:32−38aa C:60−73aa). The homeodomain (HD) is composed of 146~206 amino acids, which contains three highly conserved helical regions: helix1, helix2, and helix3 (H1 H2 H3); the nuclear localization signal (NLS) is part of H3. The COOH-terminal is composed of 238~283 amino acids, and the conserved motif (210~238 amino acids) mediates the interaction of PDX-1-PCIF1 (PDX-1 C-terminal interacting factor-1) and inhibits the transcriptional activity of PDX-1.

The human PDX-1 protein comprises 283 amino acids with a predicted molecular weight of 30.77 KDa [46]. The PDX-1 activation domain is contained within the NH2-terminal, its HD is involved in DNA binding, and they both participate in protein–protein interactions [47]. Point mutation analysis showed that the transcriptional activation region was necessary to activate insulin gene transcription [48]. The HD contains the nuclear localization signal (NLS) and an Antennapedia-like protein transduction domain (PTD), the nuclear import of PDX-1 depends on the NLS motif RRMKWKK [49]. PDX-1 mainly exists in the cytoplasm or around the nucleus in the resting state. Changes in the external environment, such as ionizing radiation and the increase in glucose concentration, can activate PDX-1 and translocate it into the nucleus [50]. The NLS of transcription factors is a crucial requirement for its action, possibly because free PDX-1 is modified by phosphorylation, acetylation, and sumoylation, which exposes the NLS and guides the nuclear translocation of PDX-1 [51–53]. In addition, Guillemain G et al. found that PDX-1 first interacts with the nuclear input receptor importinβ1 to form a complex; importinβ1 then interacts with the nuclear pore complex on the nucleus surface. It mediates PDX-1 entry into the nucleus. Ras-related nuclear protein (Ran) GTP and importinβ1 dissociate in the nucleus, followed by the reflux of the Ran GTP and importinβ1 complex into the cytoplasm through nuclear pores. Ran GTP is transformed into Ran, GDP, and the dissociated cytoplasm importinβ1 continues to participate in the transport of transcription factors [54]. In addition, it was previously reported that exogenous PDX-1 protein can permeate cells and induce insulin gene expression in pancreatic ducts because its own antennapedia-like protein transduction

domain (PTD) sequence in its structure can bind to the insulin promoter and activate its expression [55]. A conserved motif at the C-terminal of PDX-1 mediates the interaction of PDX-1-PCIF1(PDX-1 C-terminal interacting factor-1) and inhibits the transcriptional activity of PDX-1 [56]. In addition, Humphrey et al. reported a novel functional role for the PDX-1 C-terminus in mediating glucose effects. They demonstrated that glucose modulates PDX-1 stability via the AKT-GSK3 (glycogen synthase kinase 3) axis [57]. Therefore, *PDX-1* has a dual function. First, it promotes early pancreatic development and late β-cell differentiation. Second, it maintains the morphology and normal function of mature β cells, especially the normal expression of insulin secretion genes.

### 2.2. Tissue Distribution and Expression of PDX-1

*PDX-1* expression in cells at different stages is inconsistent. In the early developmental stage, *PDX-1* widely exists in the cell population transformed into endocrine and exocrine parts of the pancreas and some brain cells in the embryonic stage. However, *PDX-1* is highly expressed in β, δ, and endocrine cells of the duodenum following maturation. In contrast, its expression is low in some ductal and acinar cells [58,59]. Stoffers et al. found that *PDX-1* played an important role in developing the exocrine and endocrine portions of the pancreas, pancreatic ducts, pyloric glands of the distal stomach, common bile and cystic ducts, the intestinal epithelium of the duodenum, Brunner's glands, and the spleen [60]. In addition to the gastrointestinal system, *PDX-1* is expressed in embryonic brain cells during the active nervous system generation phase [61]. Researchers have reported *PDX-1* overexpression in various human tumor tissues [62–64]. Wang XP et al. used tissue microarray and immunohistochemical techniques to demonstrate *PDX-1* overexpression in breast, kidney, pancreatic, and prostate cancers. They suggested that *PDX-1* was probably one of the early markers of tumorigenesis [65].

### 3. Factors Regulating *PDX-1* Expression

*PDX-1* is critical for maintaining β cells, and its downregulation results in β-cell dedifferentiation. The induction of *PDX-1* expression maintains mature and functional β cells. The regulation of *PDX-1* gene expression is a complex process (Figure 2) involving nutrient substances, hormones, oxidative stress, and cytokines.

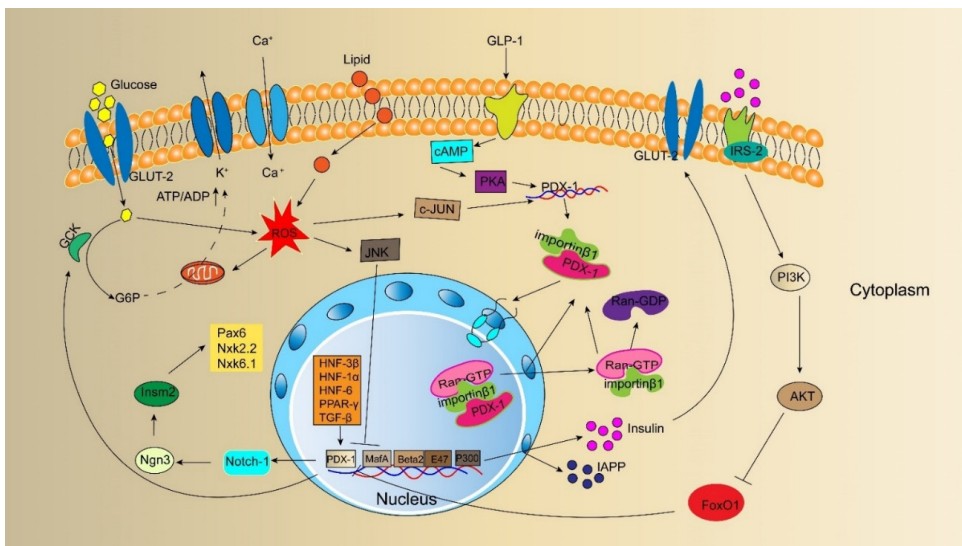

**Figure 2. Model outlining upstream regulator and direct downstream target of *PDX-1* and the insulin signaling pathway involved by *PDX-1*.** It has been identified that glucose, lipid, GLP-1, ROS, and other cytokines, such as *FoxO1*, *HNF-3β*, *HNF-6*, *PPAR-γ*, *TGF-β*, can directly regulate the expression of *PDX-1*. *PDX-1* regulates the expression of *insulin*, *GCK*, *GLUT-2*, *IAPP* to maintain β-cell characteristics and functions. In addition, *PDX-1* involves the signaling pathway of insulin secretion.

### 3.1. Nutrient Substances

Glucose and fatty acids regulate *PDX-1* expression. Many studies elucidated the underlying molecular mechanisms of glucotoxicity and lipotoxicity. In adult islet β-cell, a short-term hyperglycemic environment promotes the combination of *PDX-1* and *insulin* genes and improves *insulin* mRNA levels. However, *PDX-1* and insulin levels decrease under the cytotoxic effect of long-term hyperglycemia [66]. The inhibition of *PDX-1* expression by high glucose concentration is one of the mechanisms of glucotoxicity. Furthermore, chronic hyperglycemia has been reported to deteriorate β-cell function by inducing oxidative stress and reducing *PDX-1* DNA-binding activities [67]. In addition to activating the DNA-binding activity of *PDX-1*, glucose also affects PDX-1 phosphorylation, PDX-1 distribution between the nuclear membrane and nucleoplasm, and the transactivation potential of the amino-terminal active region of PDX-1; however, its mechanism of action remains unclear. High fatty acid concentrations also inhibit *PDX-1* expression. Gramlich et al. showed that the co-culture of pancreatic islets with palmitic acid reduced mRNA and protein expression levels of *PDX-1* by 70%. The binding force between *PDX-1*, glucose transporter 2 (*GLUT-2*), and the *insulin* gene promoter decreased by 40–65% [68]. The prolonged exposure of islets to palmitate inhibits *insulin* gene transcription by impairing the nuclear localization of PDX-1 [69]. Shimo N et al. found that *PDX-1* expression was significantly reduced during glucotoxicity and lipotoxicity. After treatment with empagliflozin or bezafibrate to selectively improve glucotoxicity or lipotoxicity, *PDX-1* showed significantly higher expression levels and enhanced β-cell proliferation [70]. The study provided further evidence that glucose and fatty acids regulate *PDX-1* expression.

### 3.2. Hormones

The incretin hormone glucagon-like peptide-1 (GLP-1) is produced by gut endocrine L cells in a nutrient-dependent manner and secreted in pancreatic islets [71]. Studies have shown that GLP-1 is involved in regulating *PDX-1*. Wang et al. found that GLP-1 promotes *PDX-1* expression in a glucose-dependent manner, increases its intracellular protein content and improves its binding activity with the A1 region of the *insulin* gene. Simultaneously, GLP-1 activates adenosine cyclase, increases cyclic adenosine monophosphate (cAMP) content in cells, and activates protein kinase A (PKA), which promotes PDX-1 synthesis and increases its content [72,73]. Hwang SL et al. found that treating rat insulinoma cells with GLP-1 significantly increased β-cell translocation gene 2 mRNA expression in dose- and time-dependent manners and subsequently elevated *PDX-1* and *insulin* mRNA levels in pancreatic β cells [74]. The activation of PKA induced by GLP-1 increases PDX-1 level and translocation to the nucleus, where *PDX-1* binds to the *insulin* gene promoter to initiate insulin expression and synthesis [75]. In addition, the runaway signal system of the small Ran GTPase significantly downregulates *PDX-1* expression in postnatal mice, resulting in insulin deficiency, decreased cell proliferation rate, and diabetes [76]. In aging animal models of type 2 diabetes, the expression of the *PDX-1* gene was decreased, the number of β cells was reduced, and the long-term use of GLP-1 reversed these pathological changes. However, when exendin (9–39) (a specific antagonist of GLP-1) was infused, its effects on the levels of *PDX-1* messenger RNA were eliminated [77]. These studies showed that GLP-1 stimulates pancreatic cell proliferation and β-cell differentiation by regulating *PDX-1*.

### 3.3. Oxidative Stress

Some studies have shown that reactive oxygen species (ROS) reduce *PDX-1* mRNA synthesis, leading to a decline in PDX-1 synthesis and a reduction in the binding of *PDX-1* to the *insulin* gene promoter and the transcription of the *insulin* gene. Cannabinoids have strong antioxidant properties. Baeeri et al. used 10 μM cannabidiol and tetrahydrocannabinol to treat aged rat islet cells. The results showed that the percentage of ROS was significantly reduced with the elevation of *PDX-1* expression and insulin release [78]. However, the results were preliminary, and further studies are needed to elucidate the mechanism. Leenders et al. treated human islets with 200 μM hydrogen peroxide for 90 min

and found that the gene and protein expression of the key transcription factor *PDX-1* was reduced by over 60% [79]. Furthermore, Matsuoka et al. treated HIT-T15 cells cultured in vitro with the oxidative stress inducer d-ribose. They found that *PDX-1* expression in cells was reduced, and the binding of *PDX-1* to the *insulin* gene promoter was significantly reduced. However, 1 mM aminoguanidine or 10 mM N-acetyl-L-cysteine (NAC) prevented the effects of d-ribose [80]. Kawamori et al. also found that oxidative stress affects the nucleocytoplasmic translocation of PDX-1 by activating c-Jun N-terminal kinase (JNK) and pointed out that JNK may induce the translocation of PDX-1 from the nucleus to the cytoplasm by activating the nuclear output signal of PDX-1, reducing the expression of *PDX-1* in the nucleus, and also reducing insulin synthesis and secretion [81,82]. After treating diabetes mice (C57BL/KsJ-db/db) with antioxidant drugs (NAC), Kajimoto et al. found that the expression of *PDX-1* in the nucleus of pancreatic islets was significantly increased, and the amounts of insulin content and *insulin* mRNA were preserved [83]. Nucleocytoplasmic translocation of PDX-1 is the key to promoting insulin secretion. Oxidative stress can affect the nucleocytoplasmic translocation of PDX-1, reduce the interaction between *PDX-1* and *insulin* gene promoters, and reduce insulin synthesis and secretion. Baumel-Alterzon S et al. reported that the nuclear factor erythroid 2-related factor (Nrf2) antioxidant pathway controls the redox balance and allows the maintenance of high PDX-1 levels; pharmacological activation of the Nrf2 pathway may alleviate diabetes by preserving PDX-1 levels [84].

### *3.4. Cytokines*

Many cytokines (hepatocyte nuclear factor (*HNF-3β*), *HNF-6*, transforming growth factor-beta (*TGF-β*), peroxisome proliferator-activated receptor gamma (*PPAR-γ*)) act upstream of *PDX-1* in the regulatory hierarchy governing pancreatic development. *HNF-3β* (also named *Foxa2*) is a transcription factor in the HNF family. *HNF-3β* binds to multiple sites in the transcriptional activation functional region of the *PDX-1* gene and recruits other transcription factors to the regulatory region to enhance gene transcription [85]. In addition, co-transfection experiments suggested that *HNF-3β*, *HNF-1α*, and specificity protein 1 (Sp1) are positive human *PDX-1* enhancer elements with mutual coordination [86]. Gao et al. found that compound conditional ablation of *Foxa1* and *Foxa2* caused near severe pancreatic hypoplasia and total loss of *PDX-1* expression, and *Foxa2* appeared to predominate. Jacquemin et al. also found that *HNF-6* acted upstream of *PDX-1* during the development of the pancreas, and *HNF-6*(-/-) mice were hypoplastic [87]. Sayo et al. used the *TGF-β* to process pancreatic β cells, and the results showed that *TGF-β* activates the *insulin* gene by activating *PDX-1* [88]. *PPAR-γ* agonist rosiglitazone was reported to increase the immunostaining of *PDX-1* and *Nkx6.1*, while *PPAR-γ* inhibitors reduced the mRNA levels of *PDX-1* through RNA interference [89]. In addition, the study found that *PDX-1* interacted with the *p300* coactivator, β-cell E-box transcription factor (*BETA2*), and *E47* coactivator to mediate *insulin* gene transcription [58]. These cytokines play a significant role in developing the pancreas; however, further studies are needed to explore their regulation to ensure *PDX-1* transcription.

## 4. *PDX-1* and the Pancreas

### *4.1. PDX-1 Promotes Pancreatic Development and Islet Cell Differentiation*

Neurogenin 3 (*Ngn3*) is critical for inducing endocrine islet cell differentiation from embryonic pancreatic progenitors, and all pancreatic endocrine cells arise from endocrine progenitors expressing *Ngn 3* [90]. *PDX-1* directly regulates *Ngn 3* expression [91]. *PDX-1* is one of the most important transcription factors for directional development and pancreas maturation. During embryonic development, *PDX-1* promotes the early development of the pancreas and β-cell differentiation and is a marker of β-cell formation and maturation.

Before embryonic development and maturation, *PDX-1* is expressed in endocrine and exocrine cells. With the development of the pancreas, *PDX-1* is only specifically expressed in 90% of β cells and a small amount (10%) of δ cells [92]. (Figure 3).

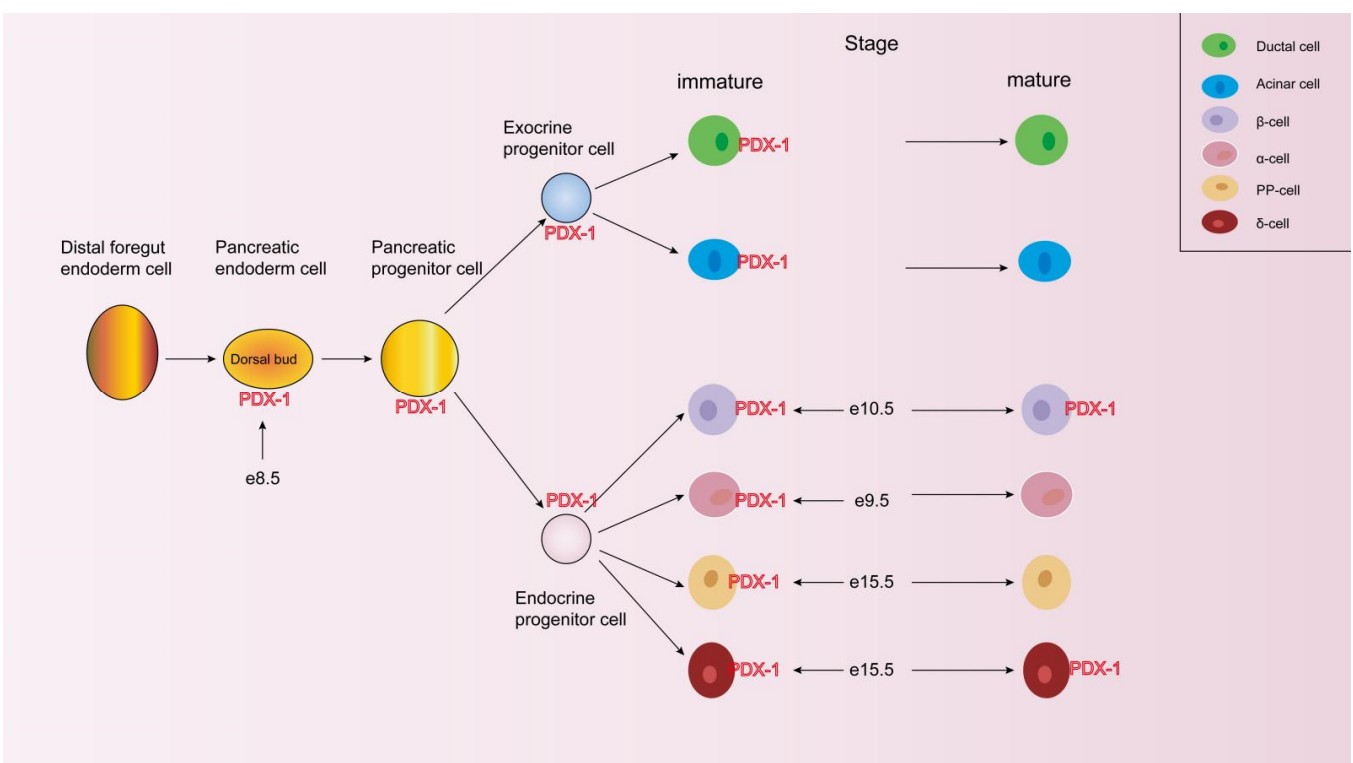

**Figure 3. Schematic representation of the *PDX-1* involved in the development of the pancreas and the differentiation of islet cells.** During embryogenesis, the pancreas arises from the foregut of the endoderm in the area that will become the duodenum. *PDX-1* is essential for pancreatic development and β-cell differentiation. It is expressed in both endocrine cells and exocrine cells before maturation and is first detected in the gut's dorsal region at embryonic days (e8.5). At e9.5, its expression is localized to the dorsal and ventral buds concomitantly with the first glucagon-secreting α cells. At e10.5, insulin-producing β-cell appear, and *PDX-1* is a marker of β-cell formation and maturation. At e15.5, *PDX-1* specifically expressed in δ cells and PP cells. With the development of the pancreas, *PDX-1* is only specifically distributed in β cells and δ cells.

Jonsson et al. reported pancreatic dysplasia in homozygous mice for a targeted mutation in the *PDX-1* gene [19]. In addition, a single nucleotide deletion in the human *PDX-1* gene coding sequence resulted in pancreatic agenesis [27,93]. Studies have shown that mice and sheep with *PDX-1* gene knockout will have pancreatic development and dysfunction, which further proves the role of *PDX-1* in early pancreatic development and the differentiation and maturation of β cells [94,95]. Wiggenhauser reported, for the first time, that *PDX-1* knockout caused the development of early diabetic kidney disease features, such as glomerular hypertrophy, filtration barrier impairments corresponding to microalbuminuria, and glomerular basement membrane (GBM) thickening in zebrafish [96]. After a partial resection of rat pancreas overexpressing the *PDX-1* gene, regenerative proliferation and repair of residual pancreatic ductal epithelial cells were observed within 24 h, further indicating that *PDX-1* promotes the differentiation of pancreatic ductal epithelial cells and pancreatic regeneration and repair [97].

### 4.2. PDX-1 Adjusts the Dedifferentiation, Redifferentiation, and Transdifferentiation of β-Cell

*PDX-1* is important in early pancreatic regulation and is indispensable for β-cell differentiation and identity maintenance. The dysfunction or failure of β-cells occurs in patients with type 2 diabetes even in the early stages of the disease. As the disease progresses, the degree of failure also increases. Previously, apoptosis was believed to be the mechanism underlying β-cell failure [98]. Recently, many studies have shown that β cells in patients with diabetes undergo a process of "dedifferentiation" rather than apoptosis.

The dedifferentiation of β cells is the central link to cell failure. In diabetes, β cells undergo dedifferentiation and return to a progenitor-like state. In 2012, Talchai et al. revealed that β-cell dedifferentiation is a key link in the pathogenesis of diabetes. They observed that in forkhead box O 1 (*FoxO1*) knockout mice, the expression of pancreatic progenitor cell markers Ngn3 and octamer-binding transcriptional factor 4 was increased, and β cells were dedifferentiated. Furthermore, the characteristics of mature cells were reversed to the state of endocrine precursor cells, which could transdifferentiate into α, PP, and other endocrine cells [99]. In another study, non-islet β cells from deceased donors were transplanted into diabetic mice. The transplanted non-islet β cells were tracked and reprogrammed by transcription factors (*PDX-1*, *MafA*) to secrete insulin and control blood glucose. The transplanted cells still produced insulin after six months [26]. Hydrodynamic gene delivery of *PDX-1*, *Ngn3*, and *MafA* to rat liver can initiate transdifferentiation to pancreatic β cells [100]. Ferber S et al. transferred the *PDX-1* gene into rat liver cells for the first time, realizing the function of insulin secretion by liver cells and causing the rat's blood glucose to decrease significantly, which was verified in the experiment of primary hepatocytes in vitro [21]. Other non-islet β cells, such as umbilical cord blood-derived mesenchymal stem cells (UCB-MSCs), bone marrow mesenchymal stem cells (BMSCs), adipose-derived mesenchymal stem cells (ADMSC), and intestinal stem cells have been reshaped into new cells whose morphology and function are similar to those of islet β cells [101]. Pham et al. reported that mRNA *PDX-1* transfection improves UCB-MSC differentiation into insulin-producing cells and produces C-peptide and insulin in a glucose-dependent manner [102].

Sun et al. constructed a eukaryotic expression vector containing *PDX-1* and transfected it into BMSCs. The results revealed that *PDX-1* promoted BMSCs to form functional pancreatic islet-like structures [103]. Gao et al. transfected the *PDX-1* gene into ADMSCs and found that it induced the differentiation of ADMSCs into functional islet-like cells [104]. In addition, pluripotent stem cell-derived pancreatic endoderm cells have been used in clinical research on type 1 diabetes [105]. These findings support the potency of *PDX*-1 in driving β-cell-like differentiation in non-β-cell, which may be a promising approach for future therapeutic use in patients with diabetes.

### 4.3. PDX-1 Regulates the Expression of the Insulin Gene and Insulin Secretion-Related Genes

*PDX-1* involves a series of gene transcriptional regulations related to maintaining β-cell characteristics and functions. *PDX-1* regulates the *insulin* gene transcription, glucokinase (*GCK*), *GLUT*-2, islet amyloid polypeptide (*IAPP*), and *somatostatin*, suggesting that *PDX-1* may play an extensive role [36]. (Figure 2).

#### 4.3.1. Insulin

*Insulin* is the most critical gene expressed explicitly in the β cells of the islet Langerhans. *PDX-1*, a homeodomain-containing transactivator of the insulin gene, plays a significant role in *insulin* gene expression and pancreatic development by binding to the enhancer region between nucleotides −340 and −91 [106]. Exposing β cells to a high-glucose environment for a short time enhanced the binding of *PDX-1* with *insulin* gene promoters and increased the level of *insulin* mRNA in vitro experiments. When β cells are exposed to high glucose for a long time, insulin gene expression defects can occur, accompanied by a decline in proinsulin mRNA levels and a reduction in islet cells, which causes a reduction in insulin synthesis. Defects in insulin gene expression are mainly due to long-term high glucose production; oxidative stress activates the JNK pathway, directly inhibits the expression of the *insulin* gene, and inhibits the activities of two main transcription factors, *PDX-1* and *MafA*. The latter two are the main participants in the activation of the *insulin* gene promoter [81,107,108]. Point mutations in the rat *insulin* gene promoter or the corresponding human sites can significantly decrease promoter activity, affecting *insulin* gene expression [109]. After silencing the *PDX-1* gene in neonatal mouse islet β cells, insulin secretion is defective, causing impaired glucose tolerance and diabetes in

adulthood. Some studies also found that following the overexpression of the *PDX-1* gene in islets, insulin secretion, and glucose tolerance were enhanced, and the expression of *MafA* and *GCK* genes, which maintain normal β-cell function, were restored in maturity-onset diabetes of the young (MODY) mice [110]. However, *PDX-1* is a necessary, but not sufficient, key regulator of insulin gene activity. Ohneda et al. reported that *PDX-1* combines with *E47*, neurogenic differentiation 1 (*NeuroD1*), and other transcription factors at the DNA binding site of the *insulin* gene promoter to form a transcription activation complex through protein–protein interactions and cooperatively activates multiple fields with the initial complex of the transcription starting point or other related coactivators to regulate the expression of the *insulin* gene [111] (Figure 4). In addition, as a nuclear architectural factor that can organize chromatin structures, the high-mobility group AT-hook 1 (HMGA1) protein plays a critical role in pancreatic beta-cell function and insulin production by regulating *PDX-1*- and *MafA*-induced transactivation of the INS gene promoter [112].

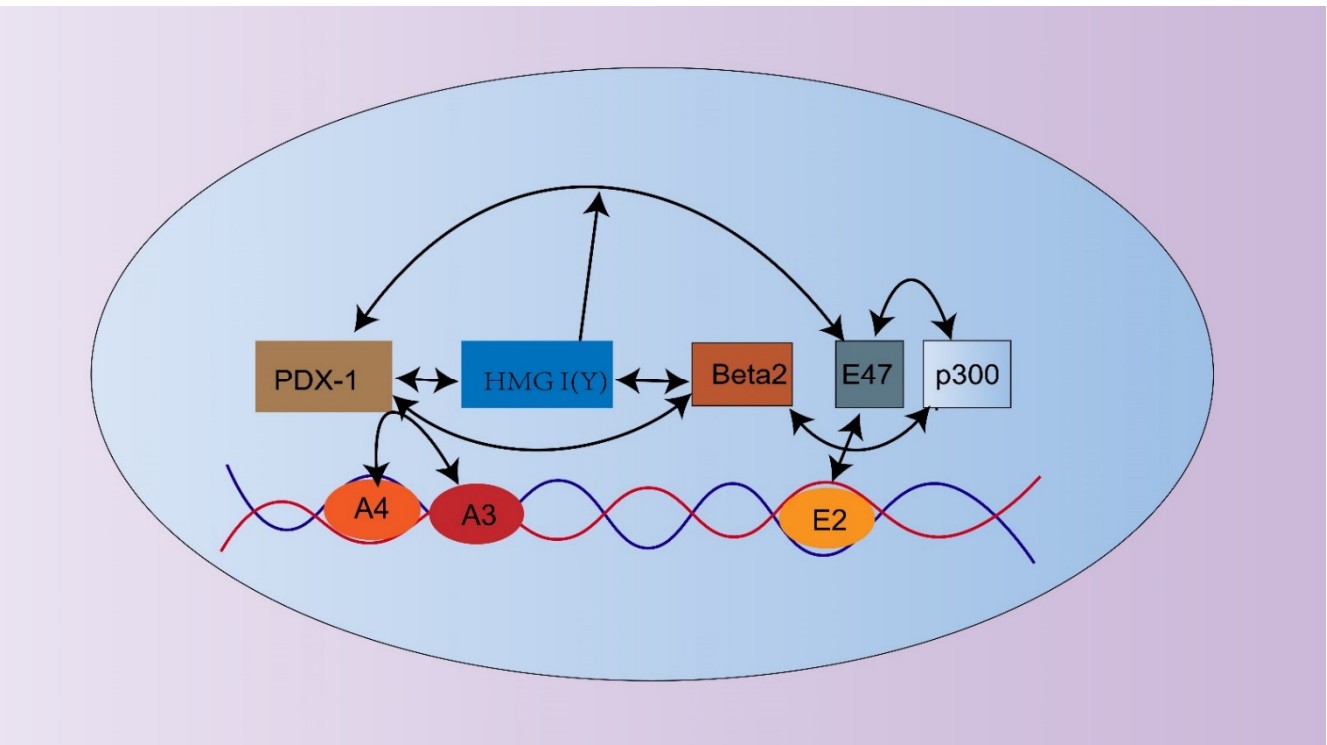

**Figure 4. The regulation of insulin expression gene and formation of transcription activation complex.** PDX-1 can bind to the A3 and A4 (collectively referred to as A3/4) sites and activate the E2A3/4 mini-enhancer by synergizing with E47 bound to the E2 site. PDX-1 and E47 do not activate the E2A3/4 mini-enhancer either individually or together. The addition of high-mobility group protein I(Y) (HMG I(Y)) allows for strong cooperative activation of the mini-enhancer by the three proteins together, but only if both the E and the A binding sites are intact in the mini-enhancer. Beta2 and HMG I(Y) contribute to PDX-1–E47 synergy through direct interactions with the homeodomain of PDX-1. The homeodomain of PDX-1 acts as a protein–protein interaction domain to recruit multiple proteins, including E47, Beta2, and HMG I(Y), to an activation complex on the E2A3/4 mini-enhancer. The p300 coactivator interacted with the activation domains of Beta2 and E47 both physically and functionally.

### 4.3.2. GCK

As the first rate-limiting enzyme, GCK serves as a glucose sensor in pancreatic β cells and is mainly concentrated in pancreatic cells, liver cells, the hypothalamus, and the gastrointestinal tract [113]. *PDX-1* is involved in regulating the *GCK* gene and is crucial in maintaining normal beta cell functions. *PDX-1* binds to the hUPE3 site (a primarily

important cis-motif of the *GCK* gene β-cell-type promoter) and activates the transcription of the *GCK* gene [114]. Li et al. found that differentiated *PDX-1* + hMSCs expressed *GCK* and released insulin/C-peptide in a weak glucose-regulated manner [115]. However, Watada et al. showed that the exogenous expression of *PDX-1* in alphaTC1 clone 6 cells alone could not induce *GCK* expression; when betacellulin was added to the medium, it expressed *insulin* and *GCK* mRNAs [116]. This study suggests that *PDX-1* can regulate the expression of *GCK*, and certain regulatory factors contribute to the β-cell specificity of gene expression.

### 4.3.3. GLUT-2

The glucose transporter GLUT-2 is expressed in the liver, intestine, kidney, pancreatic islet β cells, central nervous system, neurons, astrocytes, and hypothalamic glial cells. It is required for glucose-stimulated insulin secretion in pancreatic β cells [117]. Insulin secretion induced by glucose requires transport, storage, oxidative metabolism, and information transmission. GLUT-2 is a special transmembrane transporter in the β-cell membrane that enters cells through GLUT-2 on the membrane and is phosphorylated by GCK, causing a series of oxidative metabolic processes. In this process, the ATP/ADP ratio is increased, ATP-sensitive potassium channels are closed, cell membrane depolarization occurs, voltage-dependent $Ca^{2+}$ channels are opened, intracellular $Ca^{2+}$ is increased, and particle aggregation and exocytosis occur in coordination with other secondary messengers. GLUT-2 and GCK are essential proteins involved in this process. PDX-1 and GTIIa are central islet-specific DNA-binding proteins that regulate *GLUT-2* transcription. *PDX-1* activates the transcription and expression of *GLUT-2* by combining with a highly conserved TAAT sequence (5′-TAATA-ATAACA-3′) located in the *GLUT-2* gene promoter, affecting β-cell function [118]. Deleting the *PDX-1* gene can reduce the expression of *GLUT-2* and decrease *GLUT-2* mRNA levels in β cells and the synthesis of GLUT-2, thereby reducing insulin secretion. A study showed that AMP-activated protein kinase (AMPK) activation induced GCK and GLUT-2 downregulation by decreasing PDX-1 protein expression, and neither AICAR (AMPK agonist) nor compound C (AMPK antagonist) could alter GCK or GLUT-2 protein and mRNA expression in the absence of PDX-1 [119]. This indicated that *PDX-1* is upstream of *GLUT-2* and can directly regulate *GLUT-2* expression.

### 4.3.4. IAPP

IAPP, also known as amylin, is a 37-amino acid peptide of the calcitonin gene family isolated from amyloid deposits of non-insulin-dependent diabetic pancreas and insulinomas. In pancreatic β cells, IAPP is co-stored with insulin in secretory granules and co-released with insulin in response to various secretagogues [120]. The level of IAPP produced must be tightly controlled because its excessive secretion inhibits glucose-induced insulin secretion [121]. The amylin gene contains AT-rich sequences in its regulatory region, similar to the *insulin* gene promoter. The β-cell-specific expression of the human *IAPP* gene is principally regulated by promoter-proximal sequences [122]. Watada H et al. reported that all three typical A element-like sequences of the *IAPP* gene that matched the CT-box consensus (AT-1, −207/−202; AT-2, −154/−142; and AT-3, −88/−83) bind specifically to *PDX-1*. However, the AT-2 site may not be involved in mediating the *PDX-1*-dependent expression of *IAPP* [120]. Watada et al. established a stable alphaTC1 clone 6-derived transfectants expressing *PDX-1* and examined the changes in their gene expression patterns. The exogenous expression of *PDX-1* in alphaTC1.6 cells alone could induce *IAPP* mRNA expression in the cells [116]. These results indicated that *PDX-1* plays a principal role in defining islet β-cell expression of the *IAPP* gene.

In addition, *PDX-1* regulated the expression of other genes related to insulin secretion, such as *somatostatin*, *synaptophysin*, and paired box 4 (*Pax4*), thereby affecting β cells, stimulating insulin secretion, and changing the insulin secretion phase [123,124].

*4.4. Mitochondrial Dysfunction*

Mitochondria play a central role in coupling glucose metabolism with insulin secretion. The ability of pancreatic β cells to perform glucose-stimulated insulin secretion depends on the generation of ATP from pyruvate within the mitochondria. Concurrent with reduced mitochondrial membrane potential and cellular ATP content, impaired mitochondrial function reduces *GCK* expression, decreases insulin secretion, and β-cell apoptosis. Lee et al. found that mitochondrial dysfunction due to metabolic stress reduced *GCK* expression through *PDX-1* downregulation via the production of ROS. This caused a decrease in the association of GCK with mitochondria, resulting in pancreatic β-cell apoptosis and reduced insulin secretion [125]. The loss of *PDX-1* leads to abnormal mitochondrial morphology and function and impaired mitochondrial turnover [126]. Islets from constitutive deletion of OGT (βOGTKO) mice displayed swollen mitochondria, reduced glucose-stimulated oxygen consumption rate, ATP production, and glycolysis. *PDX-1* overexpression increased insulin content and improved mitochondrial morphology and function in βOGTKO islets [127]. In addition, Gauthier et al. reported that *PDX-1* deficiency caused mitochondrial dysfunction and defective insulin secretion in adult mice through reducing islet nuclear-encoded mitochondrial factor A(TFAM) expression [128]. However, the mechanism by which *PDX-1* regulates mitochondrial dysfunction has not been fully elucidated, and more detailed studies are needed.

**5. *PDX-1* Is Involved in the Regulation of the Insulin Signaling Pathway**

*PDX-1* mediates glucose-stimulated *insulin* gene transcription and regulates insulin secretion in normal and diabetic mice. We summarized the upstream and downstream targets of PDX-1. (Table 1) However, the signaling pathway of insulin secretion driven by *PDX-1* has not been completely elucidated, and many studies have explored possible signaling pathways. Hao et al. found that GLP-1 receptor agonists ameliorate the insulin resistance function of islet β cells via the activation of PDX-1/JAK signal transduction in C57/BL6 mice with high-fat diet-induced diabetes [129]. Kitamura et al. found that insulin and/or insulin-like growth factor regulates β-cell mass by relieving the FoxO1 inhibition of *PDX-1* expression [130]. In addition, a study showed that when the activity of the phosphoinositide 3-kinase (PI3K)/AKT/FoxO1/PDX-1 signaling pathway in islets was upregulated, PDX-1 was redistributed from the cytoplasm to the nucleus, and insulin secretion could be improved, thus ameliorating hyperglycemia [131]. Zhang et al. also found that the knockdown of G protein-coupled receptor family C group 6 subtype A (GPRC6A) or the suppression of the PI3K/AKT signaling pathway could reverse the upregulation of GPRC6A/PI3K/AKT/FoxO1/PDX-1 caused by uncarboxylated osteocalcin. Components of osteocalcin activated the FoxO1 signaling pathway to regulate *GLUT-2* expression and to improve insulin secretion disorder caused by lipotoxicity [132]. In the complex *Ngn3* transcriptional regulation network, the *Notch* signaling pathway can indirectly regulate the expression of *PDX-1* and *Ngn3* by inducing *Sox9* expression. From the regulation of the *PDX-1/Ngn3* signaling pathway, *PDX-1* is the upstream regulator of the *Ngn3* gene, which can regulate pancreatic stem cell differentiation by activating the expression of *Ngn3*. Under the synergistic action of *PDX-1* and *Ngn3*, it regulates the promoter of insm2 and activates its expression. Insm2 further differentiated into islet stem cells by upregulating the expression of *Pax6*, *Nkx2.2*, and *Nkx6.1*, which are the key genes involved in islet stem cell differentiation. The cells then continue to differentiate into mature islet β cells [133,134]. Additionally, results obtained in *PDX-1*, *Notch-1*, and *Ngn3* knockout mice models with gastric bypass surgery (GBS) suggested that GBS in db/db mice resulted in pancreatic islet regeneration through the *PDX-1/Notch-1/Ngn3* signaling pathway [135]. *PDX-1* and its related signaling pathways are essential for glucose homeostasis. The development of drugs targeting the relevant signaling pathway of *PDX-1* in insulin regulation will play a key role in treating diabetes; however, most related studies are animal experiments in the exploratory stage.

**Table 1.** The upstream and downstream targets of *PDX-1*.

|  | Upstream | Downstream |
|---|---|---|
| 1 | glucose | insulin |
| 2 | lipid | GCK |
| 3 | GLP-1 | GLUT-2 |
| 4 | ROS | IAPP |
| 5 | HNF-3β | somatostatin |
| 6 | HNF-6 | Ngn3 |
| 7 | TGF-β | Insm2 |
| 8 | PPAR-γ | synaptophysin |
| 9 | HNF-1α | Pax4 |
| 10 | Sp1 | MafA |
| 11 | HMGA1 | Nkx6.1 |
| 12 | FoxO1 |  |
| 13 | Foxa1 |  |
| 14 | Foxa2 |  |
| 15 | PKA |  |
| 16 | c-JUN |  |
| 17 | JNK |  |

## 6. *PDX-1* and the Reversal of T2DM

Based on the importance of *PDX-1* in pancreatic development and the maintenance of β cells, more research has begun to explore the application of *PDX-1* in treating diabetes. In recent years, preclinical studies have focused on the therapeutic potential of *PDX-1* in reversing type 1 diabetes. *PDX-1* is also a potential target for treating T2DM. In addition, further progress has been made in cell remodeling, gene therapy, and specific drug development.

Inducing the differentiation of non-islet cells into islet-like cells to produce insulin has shown great potential in diabetes treatment. The Ferber laboratory proposed the concept of β-cell remodeling. They transferred the recombinant adenovirus-mediated gene of *PDX-1* into the livers of BALB/C and C57BL/6 mice. They found that the massive expression of *PDX-1* promoted the expression of the *insulin* gene, which resulted in the amelioration of hyperglycemia symptoms in diabetic model mice. This study demonstrated that *PDX-1* could reprogram extra-pancreatic tissues into a β-cell phenotype, providing a possible approach to overcome the insufficiency of islet donors. However, the expression of *PDX-1* was transient, and the experiment was terminated within 8 days [21]. However, Kojima et al. used the helper-dependent adenovirus-mediated transfer of *PDX-1* to treat diabetic mice, which caused fulminant hepatitis. This may be because *PDX-1* induces the expression of trypsin and other exocrine enzymes, which causes the self-destruction of target cells. *NeuroD* (a factor downstream of *PDX*-1) and betacellulin (*Btc*) completely reversed diabetes in mice [136]. Bahrebar et al. reported that the delivery of lentiviral *PDX-1* can induce hAMSCs to generate islet-like cells and increase their own expression and islet-related genes, such as *insulin*, *Ngn3*, and *Nkx2.2* [137]. Lima et al. used a cocktail of transcription factors, *PDX-1*, *Ngn3*, *MafA*, and *Pax4*, in combination with growth factors to reprogram exocrine tissue and found that islet-like cells could be efficiently reprogrammed into islet-like cells. However, the expression of insulin protein in islet-like cells is only 15–30% of that in adult human islets, which needs further studies to resolve [138]. In addition, other non-islet β cells have been reshaped into islet-like cells [102–104]. Therefore, *PDX-1* can be used as a potential target for the clinical treatment of diabetes, and a treatment strategy based on *PDX-1* has great clinical application prospects.

In addition, *PDX-1* and *MafA* can convert islet α cells into β-like cells. Matsuoka et al. successfully constructed transgenic mice that can conditionally target the expression of MafA and *PDX-1* genes. They reported that *PDX-1* + *MafA* could transdifferentiate α cells into β-like cells by inducing key features of the β-cell and suppressing those of the α-cell [139]. Furuyama K et al. obtained α and γ cells from deceased non-diabetic or

diabetic human donors after the transduction of *PDX-1* and *MafA* with adenoviral vectors into sorted cells and reconstituted pseudo-islets. When transplanted into diabetic mice, converted human α cells reversed diabetes and continued to produce insulin even after 6 months [26]. Xiao X et al. used virus transfection to treat diabetes and found that gene therapy can reverse diabetes by converting the α cells in the pancreas into fully functioning β cells. In addition, *insulin* gene expression recovered and remained at normal levels for up to 4 months [28]. Krishnamurthy M et al. constructed organoids derived from induced pluripotent stem cells from *PDX-1*188delC/188delC patients; after the CRISPR-mediated correction of *PDX-1* point mutation, all organoid pathologies were reversed [140]. Gene therapy has had remarkable impacts in preclinical research; however, its practical clinical application is limited and requires further studies. A safer and more effective way to accurately deliver targeted genes is an urgent problem to solve.

The development of drugs that target the relevant signaling pathway of *PDX-1* in insulin regulation is crucial for diabetes treatment. There are some representative potential *PDX-1* inducing drugs/small molecules and natural compounds. (Table 2) Tu et al. reported that exendin-4 can improve T2DM progression by reversing global pancreatic histone H3K9 and H3K23 acetylation, H3K4 mono-methylation, and H3K9 di-methylation and can also reverse the inhibitory state of *PDX-1* [141]. DA-1241, a novel small-molecule G protein-coupled receptor 119 agonist, can preserve pancreatic functions by suppressing ER stress and increasing the expression of *PDX-1*; it may be a promising drug for treating diabetes [142]. Liraglutide can restore the expression of *PDX-1* and upregulate mitophagy to restore mitochondrial function and ameliorate β-cell impairment [143]. Acarbose ameliorates spontaneous type-2 diabetes in db/db mice by promoting the proliferation of islet β cells and inhibiting *PDX-1* methylation [144]. Natural compound tectorigenin (TG) can enhance the activity of the promoter of the *PDX-1* gene and activate extracellular signal-related kinase (ERK) to increase the expression of *PDX-1* [145]. Silymarin can induce the expression of *PDX-1* gene to increase serum insulin levels and decrease serum glucose levels [146]. Andrographolide, naringin, and Akebia quinata can also enhance insulin secretion and alleviate β-cell dysfunction by targeting *PDX-1* [147–149].

**Table 2.** Representative potential *PDX-1* inducing drugs, small molecules and natural compounds.

| Number | Name | Experimental Model | Administration Dose | Drug Effect | Reference |
|--------|------|--------------------|--------------------|-------------|-----------|
| 1 | Swietenine (Stn) and swietenolide (Std) | INS-1 cells (Procell CL-0368) | 2 μM, 5 μM, 8 μM, 10 μM, 15 μM, 20 μM, 30 μM, 40 μM, 50 μM | It up-regulates the expression of *PDX-1* protein, improves the insulin secretion function, protects oxidative stress injury, and reduces apoptosis. | [150] |
| 2 | Loureirin B | 3-week-old male C57BL/6J mice (14–15 g) | 45 mg/kg i.g. | It activates the AKT/PDX-1 signaling pathway. | [151] |
| 3 | *Medicago sativa* L. | Bone marrow mesenchymal stem cells (MSCs) | 50 μg/mL | It has the potential of differentiation induction of MSCs into IPCs with the characteristics of pancreatic β-like cells. | [152] |
| 4 | Nigella sativa seed | Male diabetic Wistar rats | 200 mg/kg, 400 mg/kg p.o. | It reduces oxidative stress and tissue damage, modifies the expression levels of *PDX-1* and *MafA* genes, and regulates insulin secretion and blood glucose levels. | [153] |

**Table 2.** *Cont.*

| Number | Name | Experimental Model | Administration Dose | Drug Effect | Reference |
|---|---|---|---|---|---|
| 5 | HDPs-2A (a polysaccharide purified from Hovenia dulcis) | 7-week-old male Sprague Dawley (SD) rats (170 ± 10 g) | 300 mg/kg, 200 mg/kg, 100 mg/kg, p.o. | It up-regulates *PDX-1*, activates and up-regulates IRS2 expression, and regulates apoptosis and regeneration of islet β cells to recover islet β-cell function injury in T1DM rats. | [154] |
| 6 | Thiamine disulfide (TD) | 4-week-old male Wistar rats (180–250 g) | 40 mg/kg i.p. | It increases serum insulin levels, IIR, and expression of *PDX-1* and GLUT-2 genes. | [155] |
| 7 | Gymnemic acid (GA) | 2-month-old male albino Wistar rats (130–150 g) | 150 mg/kg p.o. | It ameliorates pancreatic β-cell dysfunction by modulating *PDX-1* expression. | [156] |
| 8 | Cordycepin | INS-1 cells | 0.5~20 μM | It upregulates the mRNA level and protein expression of insulin, *PDX-1*, and GLUT-1. | [157] |
| 9 | Carnosic acid (CA) | INS-1 cells | 2.5 μM, 5 μM, 10 μM | It can protect β-cells through the PI3K/AKT/PDX-1/insulin pathway and mitochondria-mediated apoptosis. | [158] |
| 10 | Icariin | 6-week-old male albino rats (170–200 g) | 100 mg/kg p.o. | Icariin, and/or MSCs promoted the regeneration of pancreatic tissues by releasing *PDX-1* and MafA involved in the recruitment of stem/progenitor cells in the tissue. | [159] |
| 11 | Hesperidin | Male Sprague Dawley rats | 100 mg·kg$^{-1}$ p.o. | It enhances β-cell proliferation and repair and raises serum insulin levels. | [160] |
| 12 | A new form of silymarin solution (NFSM) | Male Wistar rats (220–250 g) | 100 mg/kg p.o. | It increases the expression of *PDX-1* and *insulin* genes. | [161] |
| 13 | Oligosaccharide fraction isolated from Rosa canina | 8-week-old male Wistar rats (200–250 g) | 10, 20 and 30 mg/kg i.g. | It increases the expression of *PDX-1* and may contribute to the modulation of DNA methylation. | [162] |
| 14 | Andrographolide named C1037 | 8-week-old male Kunming mice (18–22 g) | 50 mg/kg i.g. | It promotes pancreatic duct cell differentiation into insulin-producing cells by targeting *PDX-1*. | [149] |
| 15 | Tectorigenin | INS-1 cells;Diet-induced obese C57BL/6J mice | 40 μg/mL; 10, 20, 40 mg/kg i.p. | It enhances *PDX-1* expression and protects pancreatic β-cells by activating ERK and reducing ER stress. | [145] |

**Table 2.** *Cont.*

| Number | Name | Experimental Model | Administration Dose | Drug Effect | Reference |
|---|---|---|---|---|---|
| 16 | Stigmasterol-3-O-β-d-glucoside | INS-1 cells | 5 μM,10 μM | It enhances the PI3K-dependent phosphorylation of Akt at Ser473. The PI3K-dependent phosphorylation of Akt induces the movement of *PDX-1* from the nucleus to the cytoplasm and regulates the proliferation of pancreatic β-cells. | [147] |
| 17 | Naringin (4′,5,7-Trihydroxyflavanone 7-Rhamnoglucoside) | Male adult Wistar rats (250–300 g) | 100 mg/kg p.o. | It increases *insulin* gene expression and insulin secretion by upregulating the *PDX-1* gene and protein expression. | [148] |
| 18 | Small molecule kaempferol | INS-1E cells | 0.1 μM, 1 μM, 10 μM | It protects islet cells through PDX-1/cAMP/PKA/CREB signaling pathway. | [163] |
| 19 | Resveratrol | αTC9 cells | 25 μM | It inhibits histone deacetylase and promotes insulin expression synthesis by increasing *PDX-1* expression levels. | [164] |
| 20 | Jin-tang-ning (JTN) | 8-week-old female KKAy mice and gender-matched C57BL/6J mice | 8 g JTN powder/kg | It upregulates expression levels of GCK and *PDX-1*. | [165] |
| 21 | DA-1241 | 7-week-old male ICR mice and Sprague-Dawley (SD) rats | 100 mg/kg i.p. | It can preserve pancreatic functions by suppressing ER stress and increasing *PDX-1* expression. | [142] |
| 22 | Acarbose | 8-week-old db/db mice and male -+/db mice | 9 g/kg | It prevents the nuclear export of *PDX-1* and blocks the increase in methylated *PDX-1* in T2DM mouse β-cells. | [144] |
| 23 | Liraglutide | Rat RINm5F β-cell | 0.1 μmol/L | It can restore the expression of *PDX-1* and upregulate mitophagy to restore mitochondrial function and ameliorate β-cell impairment. | [143] |
| 24 | Exendin-4 | 3-week-old C57BL/6J mice | 10 μg/kg | It can improve T2DM progression by reversing global pancreatic histone H3K9 and H3K23 acetylation, H3K4 mono-methylation, and H3K9 di-methylation and also reverse the inhibitory state of *PDX-1*. | [141] |

These results suggest that *PDX-1*, a novel therapeutic target for diabetes, plays a crucial role. The expression of such a combination of transcription factors is beneficial and efficient for replacing the reduced insulin biosynthesis associated with diabetes and treating the disease.

## 7. Conclusions and Future Prospects

The treatment of diabetes is a systematic and complex process. Traditional treatment may delay disease progression, but it cannot be cured. *PDX-1*, the main transcription factor regulating islet β cells, is essential for the transcription of the *insulin* gene, insulin secretion, and proliferation and differentiation of β cells. Therefore, *PDX-1*-targeted therapy based on the relevant mechanism of *PDX-1* provides a direction for treating diabetes. However, based on existing experimental results, non-β-cell cannot secrete insulin after *PDX-1* is introduced. This shows that *PDX-1* is not the only element that transforms other cells into β cells; however, it is a crucial factor. Other essential cofactors may exist, such as *BETA*2, *NeuroD1*, and *Pax4*. It also needs the synergistic effect of various inducing conditions. In addition, the expression of *insulin* gene in islet-like cells is lower than that in adult human islets. It is also unclear whether the insulin secreted by these non-insulin-secreting cells introduced into the body fluctuates with blood glucose levels. This also explains the complexity of *insulin* gene expression regulation. However, with recent advances in translational medicine research, *PDX-1* may reverse diabetes and become a valuable tool for insulin therapy strategy and an effective target of diabetes pharmacogenomics.

**Author Contributions:** J.T. developed the review question. The initial literature review was performed by Y.Z. Y.Z. wrote the first draft of the manuscript, with X.F., J.W., R.M., H.W. and K.M. commenting on subsequent versions. All authors have read and agreed to the published version of the manuscript.

**Funding:** This work was supported by the National Natural Science Foundation of China (81904187); Capital Health Development Research Project (CD2020-4-4155); CACMS Outstanding Young Scientific and Technological Talents Program (ZZ13-YQ-026); Scientific and technological innovation project of China Academy of Chinese Medical Sciences (CI2021A01601); Innovation Team and Talents Cultivation Program of National Administration of Traditional Chinese Medicine (ZYYCXTD-D-202001); Open Project of National Facility for Translational Medicine (TMSK-2021-407).

**Acknowledgments:** We would like to thank all the authors for their contribution to the realization of this manuscript.

**Conflicts of Interest:** The authors declare that the research was conducted in the absence of any commercial or financial relationships that could be construed as a potential conflict of interest.

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
