# Peer review of "PDX-1: A Promising Therapeutic Target to Reverse Diabetes"

_biomolecules, doi:10.3390/biom12121785_

Round 1
Author Response
- A brief description of the other factors that work together with PDX-1 need to be added in the description.
Response: Thanks for your advice and we have added the relevant information.
- A table can be added summarizing the potential PDX-1 inducing drugs/small molecules and natural compounds.
Response: Thanks for your advice and we have added a summary table about the potential PDX-1 inducing drugs/small molecules and natural compounds.
- Introduction, Page 1: remove ‘s’ from carbohydrates, proteins and fats metabolism.
Response: Thanks for your advice and we have corrected it.
- Page 3: Researchers have reported PDX-1 overexpression in various human tumor tissues(53-55).
Response: Thanks for your advice and we have corrected it.
- Page 3: They suggested that PDX-1 was probably one of the early markers of tumorogenesis(56).
Response: Thanks for your advice and we have corrected it.
- In the figures, it will be preferred to use a font like Arial.
Response: Thanks for your advice and we have refined the figures.
- Page 5: Baeeri et al. used 10 mMcannabidiol and tetrahydrocannabinol to treat aged
rat islet cells.
Response: Thanks for your advice and we have corrected it.
- Page 6: Jacquemin et al. also found that HNF-6 acted upstream of PDX-1 during the development of the pancreas, and HNF-6(-/-) mice were hypoplastic(77).Error! Reference source not found. Please check such instances throughout the manuscript for references.
Response: We thank the reviewer's careful proofreading and we have checked such instances throughout the manuscript for references.
- Page 11: Matsuoka etal. successfully constructed transgenic mice that can conditionally target the expression of MafAand PDX-1 genes.
Response: Thanks for your advice and we have corrected it.
- Page 12: Naturalcompound tectorigenin (TG) TG can enhance the activity of the promoter of the PDX-1 gene and activate extracellular signal-related kinase (ERK) to increase the expression of PDX-1(132).
Response: Thanks for your advice and we have corrected it.
- Page 12: Conclusions and Future Prospects
Response: Thanks for your advice and we have corrected it.
- Page 12: The treatment of diabetes is a systematic and complex process.
Response: Thanks for your advice and we have corrected it.
- Page 12: However, based on existing experimental results, non-β-cell cannotsecrete
insulin after PDX-1 is introduced.
Response: Thanks for your advice and we have corrected it.
Reviewer 2 Report
Zhang et al. discussed about PDX-1 as a therapeutic target for diabetes reversal. Authors mentioned about the general organization of PDX-1, Its downstream and upstream targets, impact on insulin secretion and regulation of insulin signaling pathway, and PDX-1 as target to reverse Type 2 diabetes mellitus. The importance of PDX-1 in diabetes is well known but there are other factors too that also contribute to the insulin secretion as well as beta-cells transformation. Current review mainly focuses on PDX-1 as a target for diabetes reversal and there are many failed stories on using PDX-1 as a diabetes reversal. Although Authors have touched most of the aspects of PDX-1 but still I have concerns before publication. I suggest including more recent references to the manuscript with clarity in figures as well as text, and also include more information on PDX-1 regulation and interactions with other factors involved in diabetes.
1. In figure 1, it seems C-terminal of PDX-1 is also proline rich. Authors mentioned about the N-terminal in figure legend, but they did not mention anything about C-terminal part of PDX-1 in figure legend. Is there any significance of highly proline rich for PDX-1.
2. There is very little information on structure part of PDX-1 and no information at all about the important residues or missense mutations in the PDX-1 that are correlated to diabetes.
3. I also suggest summarizing various upstream and downstream targets of PDX-1 as a tabular form. Figure 2 is so confusing. in the text it is mentioned that activation of PKA increase the PDX-1 level and translocation to the nucleus, but in the figure 2 it is shown to connecting with ROS. I did not get it at all. There are many such kind of confusions in the figure 1. I suggest remaking the figure with more clarity.
4. PDX-1 is highly regulated by epigenetic signals. There is very little information on that. Such as mentioned in a current review by Liu et al., 2021. Authors mentioned that oxidation stress reduces PDX-1 mRNA synthesis and transcription of insulin gene. The information about a key regulator of the redox balance in beta cells (Nrf2) is missing. I suggest including such kind of information in the review will make it more comprehensive.
5. Authors discussed about regulation of mitochondrial dysfunction by PDX-1. I wonder why authors did not describe anything about TFAM (Gauthier et al. 2009, Cell Metabolism) that is related to mitochondrial dysfunction.
6. No information about PTD domain of PDX-1 that helps PDX-1 to permeate through cell membrane.
7. No information about HMGA1 in insulin secretion (Arcidiacono et al., 2015, Frontiers in Endocrinology).
8. Authors discuss about the regulation of insulin expression gene and formation of transcription activation complex. I think authors should make a figure to make it clearer.
9. many new references are missing those are relevant to current review. (such as Usher et al., 2022, JBC; Alterzon et al., 2022, Frontiers in Endocrinology…..)
10. There are many typos such as
Page 2: 6KB---change into 6 Kb
Page 2: molecular weight should be 31Kda rather than 31KB. Exactly it is 30.77 KDa.
Page 5: ten uM change to 10 uM.
Figure 3: It is endodem or endoderm???
Foxo1 change into FoxO1 on page 10
11. On many places: Error! Reference source not found.???? Such as page 6 and 7 and many more. Authors carefully check the text for these kinds of errors.
12. Please expand e8.5 in figure legend of figure 3. What is this about?
Author Response
- In figure 1, it seems C-terminal of PDX-1 is also proline rich. Authors mentioned about the N-terminal in figure legend, but they did not mention anything about C-terminal part of PDX-1 in figure legend. Is there any significance of highly proline rich for PDX-1?
Response: We would like to thank the reviewer for the advice and we have supplemented the relevant information. The COOH-terminal is composed of 238~283 amino acids, and the conserved motif (210~238 amino acids) mediates the interaction of PDX-1-PCIF1(PDX-1 C-terminal interacting factor-1) and inhibits the transcriptional activity of PDX-1.
- There is very little information on structure part of PDX-1 and no information at all about the important residues or missense mutations in the PDX-1 that are correlated to diabetes.
Response: We would like to thank the reviewer for the advice and we have supplemented the relevant information.
- I also suggest summarizing various upstream and downstream targets of PDX-1 as a tabular form. Figure 2 is so confusing. in the text it is mentioned that activation of PKA increase the PDX-1 level and translocation to the nucleus, but in the figure 2 it is shown to connecting with ROS. I did not get it at all. There are many such kind of confusions in the figure 1. I suggest remaking the figure with more clarity.
Response: We would like to thank the reviewer for the advice and we have summarized various upstream and downstream targets of PDX-1 in a tabular form and remade the figure with more clarity.
- PDX-1 is highly regulated by epigenetic signals. There is very little information on that. Such as mentioned in a current review by Liu et al., 2021. Authors mentioned that oxidation stress reduces PDX-1 mRNA synthesis and transcription of insulin gene. The information about a key regulator of the redox balance in beta cells (Nrf2) is missing. I suggest including such kind of information in the review will make it more comprehensive.
Response: We would like to thank the reviewer for the advice and we have supplemented the relevant information.
- Authors discussed about regulation of mitochondrial dysfunction by PDX-1. I wonder why authors did not describe anything about TFAM (Gauthier et al. 2009, Cell Metabolism) that is related to mitochondrial dysfunction.
Response: We would like to thank the reviewer for the advice and we have supplemented the relevant information.
- No information about PTD domain of PDX-1 that helps PDX-1 to permeate through cell membrane.
Response: We would like to thank the reviewer for the advice and we have supplemented the relevant information.
- No information about HMGA1 in insulin secretion (Arcidiacono et al., 2015, Frontiers in Endocrinology).
Response: We would like to thank the reviewer for the advice and we have supplemented the relevant information.
- Authors discuss about the regulation of insulin expression gene and formation of transcription activation complex. I think authors should make a figure to make it clearer.
Response: We would like to thank the reviewer for the advice and we have made a figure to make it clearer.
- many new references are missing those are relevant to current review. (such as Usher et al., 2022, JBC; Alterzon et al., 2022, Frontiers in Endocrinology…..)
Response: We would like to thank the reviewer for the advice and we have supplemented the relevant information.
- There are many typos such as
Page 2: 6KB---change into 6 Kb
Page 2: molecular weight should be 31Kda rather than 31KB. Exactly it is 30.77 KDa.
Page 5: ten uM change to 10 uM.
Figure 3: It is endodem or endoderm???
Foxo1 change into FoxO1 on page 10
Response: Thanks for your advice and we have corrected them.
- On many places: Error! Reference source not found.???? Such as page 6 and 7 and many more. Authors carefully check the text for these kinds of errors.
Response: Thanks for your advice and we have checked the text for these kinds of errors.
- Please expand e8.5 in figure legend of figure 3. What is this about?
Response: We would like to thank the reviewer for the advice and we have supplemented relevant information. “PDX-1 is essential for the pancreatic development and β-cell differentiation. It is expressed in both endocrine cells and exocrine cells before maturation and is first detected in the gut’s dorsal region at embryonic days (e8.5). At e9.5, its expression is localized to the dorsal and ventral buds concomitantly with the first glucagon-secreting α cells. At e10.5, insulin-producing β-cell appear, and PDX-1 is a marker of β-cell formation and maturation. At e15.5, PDX-1 appeared in somatostatin-secreting cells ——δ cells and pancreatic polypeptides-expressing cells——PP cells.”
Round 2
Reviewer 2 Report
The authors responded to the previous concerns and I am satisfied with that. some minor suggestions before final publication.
1. In the newly added text, fix the issue in writing sentence for reference
Such as on page 2: Three nuclease-hypersensitive sites were identified within the 5'-flanking region of the endogenous PDX-1 gene: HSS1(-2560~-1880 bp), HSS2(-1330~-800bp), and HSS3(-260~+180 bp). (38)Among them,
It should be
Three nuclease-hypersensitive sites were identified within the 5'-flanking region of the endogenous PDX-1 gene: HSS1(-2560~-1880 bp), HSS2(-1330~-800bp), and HSS3(-260~+180 bp) (38). Among them,
There are many such kind of errors in writing. Please fix those.
2. On page 2: “In a study, a child’s pancreas did not develop (pancreatic agenesis) because she was homozygous”. In this text she is referred for whom: mother or child. Please be specific.
3. Page 8: figure legend: appeared in somatostatin-secreting cells ——δ cells and pancreatic polypeptides-expressing cells — — PP cells. Please fix the writing issue.
4. Last column is empty in Table 1. Please remove that. It looks odd.
Author Response
The authors responded to the previous concerns and I am satisfied with that. some minor suggestions before final publication.
1.In the newly added text, fix the issue in writing sentence for reference
Such as on page 2: Three nuclease-hypersensitive sites were identified within the 5'-flanking region of the endogenous PDX-1 gene: HSS1(-2560~-1880 bp), HSS2(-1330~-800bp), andHSS3(-260~+180 bp). (38)Amongthem,
It should be
Three nuclease-hypersensitive sites were identified within the 5'-flanking region of the endogenous PDX-1 gene: HSS1(-2560~-1880 bp), HSS2(-1330~-800bp), and HSS3(-260~+180 bp) (38). Amongthem,
There are many such kind of errors in writing. Please fix those.
Response: We thank the reviewer's careful proofreading and we have corrected them.
2.On page 2: “In a study, a child’s pancreas did not develop (pancreatic agenesis) because she was homozygous”. In this text she is referred for whom: mother or child. Please be specific.
Response: Thanks for your advice, in this text she referred to the child, and we have corrected it.
3.Page 8: figure legend: appeared in somatostatin-secreting cells——δcells and pancreatic polypeptides-expressing cells— —PP cells. Please fix the writing issue.
Response: Thanks for your advice and we have fixed it.
4.Last column is empty in Table 1. Please remove that. It looks odd.
Response: Thanks for your advice and we have removed that.
